# Learning Unified Representations for Multi-Resolution Face Recognition

## Abstract

In this work, we propose Branch-to-Trunk network (BTNet), a novel representation learning method for multi-resolution face recognition. It consists of a trunk network (TNet), namely a unified encoder, and multiple branch networks (BNets), namely resolution adapters. As per the input, a resolution-specific BNet is used and the output are implanted as feature maps in the feature pyramid of TNet, at a layer with the same resolution. The discriminability of tiny faces is significantly improved, as the interpolation error introduced by rescaling, especially up-sampling, is mitigated on the inputs. With branch distillation and backward-compatible training, BTNet transfers discriminative high-resolution information to multiple branches while guaranteeing representation compatibility. Our experiments demonstrate strong performance on face recognition benchmarks, both for multi-resolution identity matching and feature aggregation, with much less computation amount and parameter storage. We establish new state-of-the-art on the challenging QMUL-SurvFace 1: N face identification task.

## 1 Introduction

Machine learning has advanced tremendously driven by deep learning methods, but is still severely challenged by various data specifications, such as data type, structure, scale and size, etc. For instance, face recognition (FR) is a well-established deep learning task, while the performance degrades dramatically in the testing domain that differs from the training one, influenced by factors of variance like resolution, illumination, occlusion, etc.

Most face recognition methods map each image to a point embedding in the common metric space by deep neural networks (DNNs). The dissimilarity of images can be then calculated using various distance metrics (e.g., cosine similarity, Euclidean distance, etc.) for face recognition tasks.

Recent advancements in margin-based loss (e.g., ArcFace [1], MV-Arc-Softmax [2], CurricularFace [3], etc) enhanced discriminability of the metric space, with small intra-identity distance and large inter-identity distance. However, lack of variation in training data still leads to poor generalizability. Various useful methods are utilized to mitigate this issue. The model adapts to factors of variance by augmenting datasets, whereas the large discrepancy in data distribution could potentially weaken the model's ability to extract discriminative features with the same data scale and model structure (see Section 4.3). Fine-tuning is widely used to transfer large pretrained models to new domains with different data specifications. However, this strategy requires one to store and deploy a separate copy of the backbone parameters for every single new domain, which is expensive and often infeasible.

As known, the resolutions of face images in reality may be far beyond the scope covered by the model. As the small feature maps with a fixed spatial extent (e.g., $7 \times 7$) are mapped to an embedding

with a predefined dimension (e.g., $128 - d$, $512 - d$, etc.) by a fully connected (fc) layer, input images need to be rescaled to a canonical spatial size (e.g., $112 \times 112$) before fed into the network. However, up-sampling low-resolution (LR) images introduces the interpolation error (see Section 3.1), deteriorating the recognizable ones which contain enough clues to identify the subject. Even though super-resolution methods [4–10] are widely used to build faces with good visualization, they inevitably introduce feature information of other identities when reconstructing high-resolution (HR) faces. This may lead to erroneous identity-specific features, which are detrimental to risk-controlled face recognition.

Empirically, we can divide inputs by resolution distribution and learn to operate on them via multiple models to achieve high accuracy and efficiency. However, multi-model fashion cannot be applied directly for cross-resolution recognition as representation compatibility among models need to be guaranteed [11–15].

To improve discriminability while ensure the compatibility of the metric space for multi-resolution face representation, we learn the "unified" representation by a partially-coupled Branch-to-Trunk Network (BTNet). It is composed of multiple independent branch networks (BNets) and a shared trunk network (TNet). A resolution-specific BNet is used for a given image, and the output are implanted as feature maps in the feature pyramid of TNet, at a layer with the same resolution.

Furthermore, we find that multi-resolution training can be beneficial to building a strong and robust TNet, and backward-compatible training (BCT) [11] can improve the representation compatibility during the training process of BTNet. To ameliorate the discriminability of tiny faces, we propose branch distillation in intermediate layers, utilizing information extracted from HR images to help the extraction of discriminative features for resolution-specific branches.

Our method is simple and efficient, which breaks the convention of up-sampling the inputs and serves as a general framework that can be easily implemented by several existing methods due to conceptual simplicity. Meanwhile, BTNet is able to reduce the number of FLOPS by operating the inputs without up-sampling, and per-resolution storage cost by only storing the learned branches and resolution-aware BNs [16], while re-using the copy of the trunk model.

We demonstrate that our method performs comparably in various open-set face recognition tasks (1:1 face verification and 1: N face identification), in both settings of multi-resolution identity matching and feature aggregation, while meaningfully reduces the redundant computation cost and parameter storage. In the challenging QMUL-SurvFace 1: N face identification task [17], we establish new state-of-the-art by outperforming prior models. Furthermore, by avoiding the ill-posed problem (i.e., image up-sampling), our approach also effectively reduces the additional noise and uncertainty of the representation, which plays a key role in reliable risk-controlled face recognition.

## 2    Related Work

**Compatible Representation Learning:**    The task of compatible representation learning aims at encoding features that are interoperable with the features extracted from other models. Shen et. al. [11] first formulated the problem of backward-compatible learning (BCT) and proposed to utilize the old classifier for compatible feature learning. Since the multi-model fashion benefits representation learning with lower computation, our idea of cross-resolution representation learning can be modeled similar to cross-model compatibility [11–15], as metric space alignment for different resolutions. Our goal is achieved by both compatibility-aware network architecture and training strategy.

**Knowledge Distillation and Transfer:**    The concept of knowledge distillation (KD) was first proposed by Hinton et. al. in [18], which can be summarized as employing a large parameter model (teacher) to supervise the learning of a small parameter model (student). Distillation from intermediate features [19–29] is widely adopted to enhance the effectiveness of knowledge transfer. However, due to the "dark knowledge" hidden in the intermediate layers, additional subtle design is often required to match and rescale intermediate features. Instead, our approach can easily locate the distillation features without rescaling and effectively transfer knowledge from the HR domain to LR branches.

**Low Resolution Face Recognition:** Its task includes low resolution-to-low resolution (LR-to-LR) matching and low resolution-to-high resolution (LR-to-HR) matching [30]. The work can be divided into two categories [31]: (1) Super-resolution (SR) based methods aim to upscale LR images to construct HR images and use them for feature extraction [4–10]. (2) Projection-based methods aim to extract adequate representations in different domains and project them into a common feature space [32–34]. SR approaches are able to build faces with good visualization, but inevitably introduce feature information of other identities when reconstructing corresponding HR faces, thus introducing noise for identity-specific features. Compared to previous projection methods, our approach directly learns discriminative representations in a common feature space for HR and LR inputs, without additional projection heads for feature transformation.

**Pseudo-Siamese Networks:** Siamese networks are a coupling architecture based on DNNs, which are widely used for signature verification [35], face verification [36, 37], tracking [38], etc. Pseudo-Siamese networks [39] are decoupled Siamese networks, as the weights of the two branches are not shared, resulting in a more flexible representation way for the two entities. Hughes et. al. in [40] proposed a pseudo-Siamese CNN for identifying corresponding patches in SAR and optical images. Inspired by pseudo-Siamese networks, we propose a resolution-adaptive partially coupled Siamese network architecture, extracting specific-shared features for images with different resolutions.

# 3 Learning Specific-Shared Feature Transfer

Instead of rescaling the inputs to a canonical size, we build multiple resolution-specific branches (BNets) that are used to map inputs to intermediate features with the same resolution and a resolution-shared trunk (TNet) to map feature maps with different resolutions to a high-dimension embedding. We gain several important properties by doing so: (1) Processing inputs on its original resolution can diminish the inevitably introduced error via up-sampling or information loss via down-sampling, thus preserving the discriminability of visual information with different resolutions. (2) Information streams of different resolutions are encoded uniformly, thus enabling the representation compatibility, which is particularly beneficial to open-set face recognition considering that a compatible metric space is the prerequisite for computing similarity. (3) This also effectively reduce the computation for LR images by supplying computational resources conditioned on the input resolution.

## 3.1 Up-Sampling Error Analysis

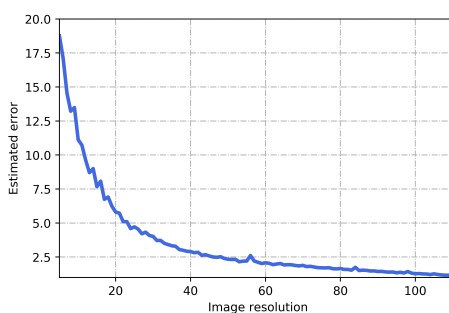

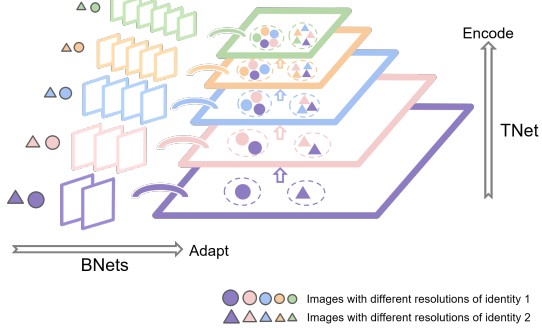

Figure 1: Estimated Error Upperbound. (bilinear interpolation, average value for over 100 images) with the change of image resolution relative to resolution 112.

Figure 2: Basic ideas of the proposed BTNet. Images of a certain identity are first projected to the feature maps with the same resolution respectively (Adapt) and then projected to a unified feature representation (Encode). In this figure, feature maps with the same resolution are indicated by outlines in the same color.

Figure 1 illustrates the experimental estimation of interpolation error, whose upper bound increases with the decline of the image resolution (see detailed theoretical derivation in Appendix A.1). Note that the error soars up when the resolution drops below 32 approximately which can be viewed as LR face images, consistent with the tiny-object criterion [41].

The results show that: (1) inputs with a resolution higher than around 32 can be considered in the same HR domain, since the error information introduced by up-sampling via interpolation can be ignored to a certain extent; (2) inputs with a resolution lower than around 32 should be treated as in various LR domains due to the high sensitivity of the resolution to errors.

## 3.2 Branch-to-Trunk Network

Let $X$ be an input RGB image with a space shape: $X \in \mathbb{R}^{H \times W \times 3}$ where $H \times W$ corresponds to the spatial dimension of the input. For efficient batch training and inference, we predefine a canonical size $S \times S$ (e.g., $112 \times 112$ for typical face recognition models like ArcFace [1]).

We build a trunk network $T : \mathbb{R}^{H \times W \times 3} \to \mathbb{R}^{C_{emb}}$ capable of extracting discriminative information with different resolutions, where $C_{emb}$ is the number of embedding channels. For every resolution $r$ in the candidate set, we formulate a resolution-specific branch, $z_r = B_r(X_r)$, which maps the input image $X_r$ to feature maps with the same resolution and expanded channels $z_r : \mathbb{R}^{r \times r \times 3} \to \mathbb{R}^{r \times r \times C_r}$. The idea is to learn our branches $B$ to focus on resolution-specific feature transfer independently. Feature maps will then be coupled to the trunk network $T$ in the feature pyramid with the same spatial resolution $r \times r$, allowing for further mapping to the unified presentation space by $T_r : \mathbb{R}^{r \times r \times C_r} \to \mathbb{R}^{C_{emb}}$.

Here, we follow the idea of "avoiding redundant up-sampling". Our branches $B$ are implemented with same-resolution mapping: i.e., the model preserves the network architecture of $T$ from input to the layer with resolution $r$ and abandons down-sampling operations (e.g., replacing the convolution of stride 2 with stride 1, abandoning the pooling layers, etc.) to keep the same-resolution flow.

We specifically name our specific-shared feature transfer network as Branch-to-Trunk Network, abbreviated as "BTNet". Figure 2 visually summarizes the main ideas of BTNet.

## 3.3 Training Objectives

We now describe the training objectives. The training of BTNet includes training the trunk network $T$ such that it can produce discriminative and compatible representations for multi-resolution information, and fine-tuning the branch networks $B$ to encourage them to learn resolution-specific feature transfer, so as to improve accuracy without compromising compatibility.

**Influence Loss.** It is a compatibility-aware classification loss which is implemented by feeding the embeddings of the new model to the classifier of the old model [11]. Since the difficulties of samples vary due to image resolution, we compute CurricularFace [3] as our classification loss, in the form of:

$$L_{cur} = -\log\Big(\frac{e^{s\cos(\theta_{y_i}+m)}}{e^{s\cos(\theta_{y_i}+m)} + \sum_{j=1, j \neq y_i}^{n} e^{sN(t^{(k)}, \cos(\theta_j))}}\Big) \tag{1}$$

$$N(t, \cos\theta_j) = \begin{cases} \cos(\theta_j), & \cos(\theta_{y_i}+m) - \cos(\theta_j) \geq 0 \\ \cos(\theta_j)(t + \cos(\theta_j)), & else \end{cases} \tag{2}$$

$$t^{(k)} = \alpha \sum_i \cos\theta_{y_i} + (1-\alpha)\, t^{(k-1)} \tag{3}$$

which distinguishes both the difficultness of different samples in each stage and relative importance of easy and hard samples during different training stages. Thus, we refine CurricularFace loss as our influence loss:

$$L_{influence} = L_{cur}(\varphi_{bt}, \kappa^*) \tag{4}$$

where $\varphi_{bt}$ is BTNet backbone (both $B_r$ and $T_r$), and $\kappa^*$ is the classifier of the pretrained trunk $T$.

**Branch Distillation Loss.** Due to the continuity of the scale change of both the image pyramid and the feature pyramid [42], we can get a qualitative sense of the similarity between images and feature maps with the same resolution (see Figure 3). Furthermore, features extracted from HR images have richer and clearer information than those from LR images [43]. Motivated by these analyses, we utilize an MSE loss to encourage the branch output $z_r$ to be similar to the corresponding feature maps of the pretrained trunk network $z_s$:

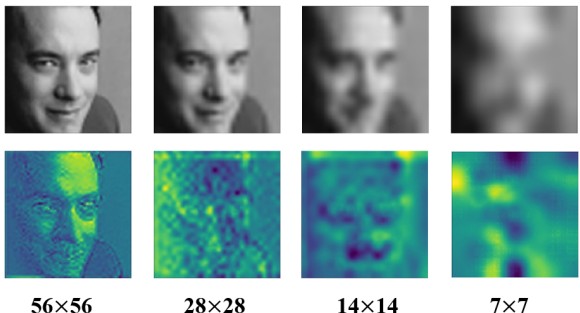

| 56×56 | 28×28 | 14×14 | 7×7 |

Figure 3: Visual comparison of face image-feature map pairs with different resolutions (resized to a common size here for illustration).

$$L_{branch} = \frac{1}{V} \sum_{v=1}^{V} \left( z_{r_v} - z_{s_v} \right)^2 \tag{5}$$

where $V$ denotes the batch size.

The whole training objective is a combination of the above objectives:

$$L = L_{influence} + \lambda_{branch} L_{branch} \tag{6}$$

where $\lambda_{branch}$ is a hyper-parameter to weigh the losses and we set $\lambda_{branch} = 0.5$ in all our experiments.



Figure 4: Comparison of # Params (M) between fully finetuning and $\varphi_{bt}$.

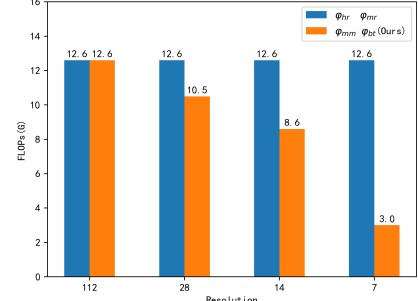

Figure 5: Comparison of FLOPs (G) between baselines and $\varphi_{bt}$.

### 3.4 Storing Branch Networks

An obvious adaptation strategy is fully finetuning of the model on each resolution. However, this strategy requires one to store and deploy a separate copy of the backbone parameters for every resolution, which is an expensive proposition and difficult to expand into more segmented resolution branches. Our BTNet is beneficial in the scenario of multi-resolution face recognition which achieves better parameter/accuracy trade-offs. Since activation statistics including means and variances under different resolutions are incompatible [44], we update and store Batch Normalization (BN) [45] parameters in all layers of $B_r$ and $T_r$ for each resolution, whose amount is negligible. Apart from this, we only need to store the learned branches and re-use the original copy of the pretrained trunk model, significantly reducing the storage cost. Figure 4 shows that BTNet requires only $1.1\% \sim 48.9\%$ of all the parameters compared to fully updating all the parameters of TNet.

## 4 Experiments

To validate BTNet on face recognition tasks in open universe, we perform 1:1 verification and $1 : N$ identification tasks in two different settings, including (a) multi-resolution identity matching, and

186 (b) multi-resolution feature aggregation. For 1:1 verification, a pair of templates are provided and
187 the model is to decide whether they belong to the same identity or not. For 1:N identification, a set
188 of gallery images are first mapped onto their embedding vectors (indexing) and the embeddings of
189 query images are extracted to perform search against indexed gallery.

## 4.1 Implementation Details

191 **Datasets.** We use MS1Mv3 [46] for training face embedding models. The MS1Mv3 dataset
192 contains 5,179,510 images of 93,431 celebrities. According to the test setting, different test datasets
193 are used.

194 **·Multi-Resolution Identity Matching.** We try on six widely adopted face verification benchmarks:
195 LFW [47], CFP-FF [48], CFP-FP [48], AgeDB-30 [49], CALFW [50], and CPLFW [51], while
196 the large-scale surveillance face dataset QMUL-SurvFace [17] is used for 1:N face identification,
197 which contains native LR surveillance faces across wide space and time. The spatial resolution for
198 QMUL-SurvFace ranges from 6/5 to 124/106 in height/width with an average of 24/20.

199 **·Multi-Resolution Feature Aggregation.** We adopt a top challenging benchmark IJB-C [52], which
200 has around 130k images from 3,531 identities, for two standard testing protocols: $1:1$ verification
201 and 1:N identification.

202 **Training.** All the models are trained on four RTX 2080 Tis with batch size 128 by stochastic
203 gradient descent. For TNet, we train for 25 epochs, with learning rate initialized at 0.2 with 2 warm-
204 up epochs and decaying as a quadratic polynomial. We augment training samples by random horizonal
205 flipping and multi-resolution training. For BNets, we initialize the learning rate by $0.02$ without
206 warm-up epochs. The training all stops at the $10th$ epoch for a fair comparison. The recommended
207 hyper-parameters are used for classification loss from the original paper (e.g., $m = 0.5, s = 64$
208 for ArcFace [1], and $\alpha = 0.99, t^0 = 0$ for CurricularFace [3]). Only horizonal flipping is used as
209 augmentation when training BNets.

210 **Baselines.** In our experiment, several baselines are used to validate BTNet in learning discriminative
211 and compatible representations for multi-resolution face recognition.

212 **·High-Resolution Trained** $\varphi_{hr}$**.** Naive baseline trained with HR data.

213 **·Independently Trained** $\varphi_{mm}$**.** Multi-model fashion: is it possible to achieve better results if we
214 train a specific model for each resolution independently? Specifically, we train $\varphi_r$ for data with
215 resolution $r$ and denote the multi-model collections as $\varphi_{mm}$.

216 **·Multi-Resolution Trained** $\varphi_{mr}$**.** Trained with multi-resolution data which adapts to resolution-
217 variance. Specifically, each image is randomly down-sampled to a size in the candidate set $\{\frac{112}{2^i} \times$
218 $\frac{112}{2^i} | i = 0, 1, 2, 3, 4\}$ with equal probability of being chosen, and then up-sampled back to $112 \times 112$.

219 **Instantiation of Network Architecture.** The BTNet and baselines are implemented with ResNet50
220 [53], and they could be extended easily with other implementations. Dubbed as $\varphi_{bt}$, the detailed
221 instantiation of BTNet based on ResNet50 is illustrated in Appendix A.2.

## 4.2 Evaluation Metrics

223 On the benchmarks for face verification, we use 1:1 verification accuracy as the basic metrics. The
224 rank-20 true positive identification rates (TPIR20) at varying false positive identification rates (FPIR)
225 and AUC are used to report the identification results on QMUL-SurvFace. The evaluation metrics
226 for IJB-C 1:1 verification protocol are true acceptance rates (TAR) at different false acceptance rate
227 (FAR). For 1:N identification, the basic evaluation metrics are the true positive identification rates
228 (TPIR) at different false positive identification rates (FPIR).

229 For better evaluation, we define another two metrics to assess the relative performance gain similar to
230 [11, 14].

Table 1: Comparison of different methods on six face verification benchmarks. "Acc." denotes average 1:1 verification accuracy.

(a) Cross-resolution identity matching.

| | 112&7 | | 112&14 | | 112&28 | |
|---|---|---|---|---|---|---|
| | Acc. | Gain | Acc. | Gain | Acc. | Gain |
| $\varphi_{hr}$ | 57.75 | - | 81.02 | - | 95.90 | - |
| $\varphi_{mm}$ | 50.58 | -0.89 | 49.90 | -4.82 | 50.03 | -305.80 |
| $\varphi_{mr}$ | 65.85 | +1.00 | 87.47 | +1.00 | 96.05 | +1.00 |
| $\varphi_{bt}$ (Ours) | 86.10 | +3.50 | 94.08 | +2.02 | 96.65 | +5.00 |

(b) Same-resolution identity matching.

| | 7&7 | | 14&14 | | 28&28 | | 112&112 | |
|---|---|---|---|---|---|---|---|---|
| | Acc. | Gain | Acc. | Gain | Acc. | Gain | Acc. | Gain |
| $\varphi_{hr}$ | 60.70 | - | 73.88 | - | 93.58 | - | **97.68** | - |
| $\varphi_{mm}$ | 62.57 | +1.00 | 78.00 | +1.00 | 94.68 | +1.00 | **97.68** | - |
| $\varphi_{mr}$ | 61.02 | +0.17 | 80.32 | +1.56 | 95.12 | +1.40 | 97.25 | - |
| $\varphi_{bt}$ (Ours) | **77.78** | **+9.13** | **90.90** | **+4.13** | **96.27** | **+2.45** | 97.25 | - |

**Cross-Resolution Gain.** With the purpose towards the cross-resolution compatible representations, we define the performance gain as follows:

$$Gain_{r_1 \& r_2}(\varphi) = \frac{M_{r_1 \& r_2}(\varphi) - M_{r_1 \& r_2}(\varphi_{hr})}{|M_{r_1 \& r_2}(\varphi_{mr}) - M_{r_1 \& r_2}(\varphi_{hr})|} \tag{7}$$

Here $M_{r_1 \& r_2}(\cdot)$ are metrics when the resolutions of the image/template pair are $r_1 \times r_1$ and $r_2 \times r_2$ ($r_1 \neq r_2$), respectively. $\varphi_{mr}$ shares the same architecture with $\varphi_{hr}$ while is trained on multi-resolution images and thus serves as the baseline of cross-resolution gain.

**Same-Resolution Gain.** For the scenario of multi-resolution face recognition, the performance of same-resolution verification/identification is also vital besides cross-resolution one. Therefore, we report the relative performance improvement from base model $\varphi_{hr}$ in the scenario of same-resolution.

$$Gain_{r \& r}(\varphi) = \frac{M_{r \& r}(\varphi) - M_{r \& r}(\varphi_{hr})}{|M_{r \& r}(\varphi_r) - M_{r \& r}(\varphi_{hr})|} \tag{8}$$

Here $M_{r \& r}(\cdot)$ are metrics when the resolutions of the image/template pair are both $r \times r$. $\varphi_r$ is a model of the set $\{\varphi_{mm} = \varphi_r | r = 7, 14, 28\}$ trained on images with resolution $r \times r$ without considering cross-resolution representation compatibility, which serves as the baseline of same-resolution gain on resolution $r$. Note that for both metrics we add the absolute symbol to the denominator as they can be negative in some test settings (detailed in Section 4.3 and 4.4).

### 4.3 Multi-Resolution Identity Matching

We now conduct experiments on the proposed BTNet framework for multi-resolution identity matching. Two different settings are included : (1) same-resolution matching, and (2) cross-resolution matching. Table 1 compares the average performance on popular benchmarks for $\varphi_{hr}$, $\varphi_{mm}$, $\varphi_{mr}$, $\varphi_{bt}$. The experimental results on each dataset are detailed in Appendix A.5.

When directly applied to test data with the resolution lower than training data, $\varphi_{hr}$ suffers a severe performance degradation. Up-sampling images via interpolation can increase the amount of data but not the amount of information, only to improve the detailed part of the image and the spatial resolution (size) [64]. Moreover, it also brings various noise and artificial processing traces [65]. Up-sampling images via interpolation-typically bilinear interpolation or bicubic interpolation of 4x4 pixel neighborhoods, essentially a function approximation method, is bound to introduce error information (detailed in Appendix A.1), thus potentially confusing identity information, which is especially crucial for LR images with limited details. We are able to observe improvement of $\varphi_{mm}$ in same-resolution matching but its cross-resolution gain is negative with approximately 50% accuracy. Unsurprisingly, independently trained $\varphi_r$ is unaware of representation compatibility, and thus does not naturally suitable for cross-resolution recognition. The results show that $\varphi_{mr}$ improved both cross-resolution and same-resolution accuracy by a large margin, as it learns to adapt to resolution variance and maintain discriminability of multi-resolution inputs. Note that the model size and training data scale stay the same, while only the resolution distribution of the data changes for $\varphi_{mr}$, and thus there is a marginal accuracy drop in the setting of 112&112 matching. Comparably, $\varphi_{bt}$ substantially outperforms all baselines with 2.02 ~5.00 cross-resolution gain and 2.45~9.13 same-resolution gain. Importantly, due to the multi-resolution branches, our approach has a cost same with $\varphi_{mm}$, significantly lower than $\varphi_{hr}$ and $\varphi_{mr}$ (see Figure 5).

Table 2: Performance of face identification on QMUL-SurvFace. Most compared results are cited from [17, 54] except BTNet.

|  | TPIR20(%)@FPIR | | | | |
|---|---|---|---|---|---|
|  | AUC | 0.3 | 0.2 | 0.1 | 0.01 |
| VGG-Face [55] | 14.0 | 5.1 | 2.6 | 0.8 | 0.1 |
| DeepID2 [56] | 20.8 | 12.8 | 8.1 | 3.4 | 0.8 |
| FaceNet [57] | 19.8 | 12.7 | 8.1 | 4.3 | 1.0 |
| SphereFace [58] | 28.1 | 21.3 | 15.7 | 8.3 | 1.0 |
| SRCNN [59] | 27.0 | 20.0 | 14.9 | 6.2 | 0.6 |
| FSRCNN [60] | 27.3 | 20.0 | 14.4 | 6.1 | 0.7 |
| VDSR [61] | 27.3 | 20.1 | 14.5 | 6.1 | 0.8 |
| DRRN [62] | 27.5 | 20.3 | 14.9 | 6.3 | 0.6 |
| LapSRN [63] | 27.4 | 20.2 | 14.7 | 6.3 | 0.7 |
| ArcFace [1] | 25.3 | 18.7 | 15.1 | 10.1 | 2.0 |
| RAN [54] | 32.3 | 26.5 | 21.6 | 14.9 | **3.8** |
| BTNet (avg.+floor) | 32.6 | 27.9 | 23.4 | 16.5 | 1.4 |
| BTNet (avg.+near) | 34.6 | 30.3 | 25.7 | 18.9 | 1.5 |
| BTNet (avg.+ceil) | **35.4** | 31.1 | 26.8 | 20.3 | 2.2 |
| BTNet (min+floor) | 32.3 | 27.6 | 23.2 | 16.1 | 1.4 |
| BTNet (min+near) | 34.0 | 29.6 | 25.0 | 18.0 | 1.4 |
| BTNet (min+ceil) | 35.3 | 31.0 | 26.6 | 19.9 | 2.0 |
| BTNet (max+floor) | 33.6 | 29.1 | 24.5 | 17.6 | 1.3 |
| BTNet (max+near) | 35.2 | 31.0 | 26.4 | 19.6 | 1.7 |
| BTNet (max+ceil) | **35.4** | **31.2** | **26.9** | **20.6** | 2.5 |

For inference on inputs with resolutions not strictly matched to the branch, we validate three selection strategies based on three resolution indicators (see Figure 6). Table 2 compares BTNet against the state-of-the-arts models on QMUL-SurvFace 1:N identification benchmark. We are able to observe that our proposed approach extends the state-of-the-arts while being more computationally efficient. We believe the performance of BTNet (max + ceil) is the highest that have been reported so far, and we believe it is meaningful with the increased focus on unconstrained surveillance applications.

## 4.4 Multi-Resolution Feature Aggregation

Multi-resolution feature aggregation is common in set-based recognition tasks where the model needs to determine the similarity of sets (templates), instead of images. Each set could contain images of the same identity with different resolutions. In our experiment, we rescale the original and flipped images in each set to different resolutions and aggregate their features into a representation of the template. Detailed experimental results can be seen in Appendix A.5.

Table 3 (a) compares the cross-resolution results of TAR@FAR=$10^{-4}$ for 1:1 verification. The cross-resolution features are ensured to be mapped to the same vector space where the aggregation is conducted for $\varphi_{hr}$ and $\varphi_{mr}$, but we can observe that $\varphi_{hr}$ performs much better than $\varphi_{mr}$. One possible reason is that $\varphi_{hr}$ has outstanding discriminability to extract HR features, while LR features may not overly deteriorate the HR information. This phenomenon also suggests that $\varphi_{mr}$ sacrifices its discriminability in exchange for the adaptability for resolution-variance. We can see $\varphi_{bt}$ is comparable with $\varphi_{hr}$, demonstrating the discriminative power of BTNet for aggregating multi-resolution features.

Table 3 (b) compares the same-resolution results of TAR@FAR=$10^{-4}$ for 1:1 verification. When HR information is removed from the template representation (i.e., test settings 7&7, 14&14, 28&28), $\varphi_{hr}$ suffers from performance degradation as well, as the informative embedding cannot catch the lost details of the LR images [54]. Both $\varphi_{mm}$ and $\varphi_{mr}$ improve with a limited same-resolution gain, while $\varphi_{bt}$ surpasses the baselines by a large margin while also reducing the compute.

In Table 4 we show the results of TPIR@FPIR=$10^{-1}$ for 1:N identification protocol. Similar to our results for 1:1 verification, we are able to observe that $\varphi_{bt}$ is comparable or even better than $\varphi_{hr}$ with

Table 3: Comparison of different methods on the IJB-C dataset 1:1 face verification task. "TAR" denotes TAR (%@FAR=1e-4).

(a) Cross-resolution feature aggregation.

| | 112&7 | | 112&14 | | 112&28 | |
|---|---|---|---|---|---|---|
| | TAR | Gain | TAR | Gain | TAR | Gain |
| $\varphi_{hr}$ | **88.89** | - | 92.40 | - | **95.62** | - |
| $\varphi_{mm}$ | 74.54 | -0.56 | 93.52 | +1.33 | 95.42 | -0.69 |
| $\varphi_{mr}$ | 63.11 | -1.00 | 91.56 | -1.00 | 95.33 | -1.00 |
| $\varphi_{bt}$ (Ours) | 88.17 | -0.03 | **93.97** | **+1.87** | 95.62 | **+0.00** |

(b) Same-resolution feature aggregation.

| | 7&7 | | 14&14 | | 28&28 | | 112&112 | |
|---|---|---|---|---|---|---|---|---|
| | TAR | Gain | TAR | Gain | TAR | Gain | TAR | Gain |
| $\varphi_{hr}$ | 4.83 | - | 33.74 | - | 89.65 | - | **96.40** | - |
| $\varphi_{mm}$ | 4.83 | + 0.00 | 29.26 | -1.00 | 92.58 | +1.00 | **96.40** | - |
| $\varphi_{mr}$ | 4.48 | - | 40.51 | +1.51 | 92.81 | +1.08 | 96.06 | - |
| $\varphi_{bt}$ (Ours) | 35.47 | - | 82.08 | +10.79 | 94.50 | +1.66 | 96.06 | - |

Table 4: Comparison of different methods on the IJB-C dataset 1: N face identification task. "TPIR" denotes TPIR (%@FPIR=0.1).

(a) Cross-resolution feature aggregation.

| | 112&7 | | 112&14 | | 112&28 | |
|---|---|---|---|---|---|---|
| | TPIR | Gain | TPIR | Gain | TPIR | Gain |
| $\varphi_{hr}$ | **85.60** | - | 90.11 | - | 94.27 | - |
| $\varphi_{mm}$ | 69.70 | -0.55 | 91.73 | +1.53 | 94.13 | -0.33 |
| $\varphi_{mr}$ | 56.64 | -1.00 | 89.05 | -1.00 | 93.84 | -1.00 |
| $\varphi_{bt}$ (Ours) | 83.93 | -0.06 | **91.87** | **+1.66** | **94.33** | **+0.14** |

(b) Same-resolution feature aggregation.

| | 7&7 | | 14&14 | | 28&28 | | 112&112 | |
|---|---|---|---|---|---|---|---|---|
| | TPIR | Gain | TPIR | Gain | TPIR | Gain | TPIR | Gain |
| $\varphi_{hr}$ | 3.12 | - | 26.37 | - | 86.06 | - | **95.57** | - |
| $\varphi_{mm}$ | 3.24 | +1.00 | 21.84 | -1.00 | 89.76 | +1.00 | 95.57 | - |
| $\varphi_{mr}$ | 3.25 | +1.08 | 37.58 | +2.47 | 91.02 | +1.34 | 94.85 | - |
| $\varphi_{bt}$ (Ours) | 27.70 | +204.83 | 76.65 | +11.10 | 92.89 | +1.85 | 94.85 | - |

HR information involved and can preserve superior discriminability with limited LR information, while also being more computationally efficient.

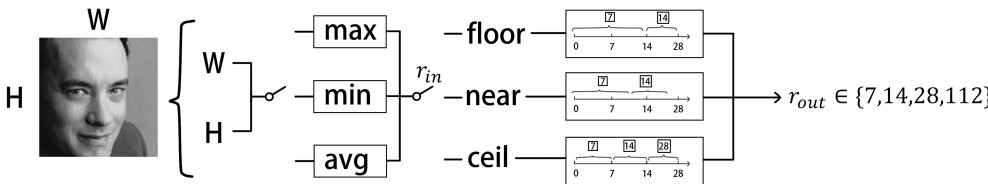

Figure 6: Branch selection process. Max/min/average is used on (W, H) to obtain a resolution indicator for further allocation (floor/near/ceil) to a certain branch.

## 5 Discussion and Conclusion

This paper works on the problem of multi-resolution face recognition, and provides a new scheme to operate images conditioned on its input resolution without large span rescaling. The error introduced by up-sampling via interpolation is investigated and analyzed. Decoupled as branches for discriminative representation learning and coupled as the trunk for compatible representation learning, our Branch-to-Trunk Network (BTNet) achieves significant improvements on multi-resolution face verification and identification tasks. Besides, the superiority of BTNet in reducing computational cost and parameter storage cost is also demonstrated. It is worth noting that our approach is easy to expand to recognition tasks for other classes of objects and has the potential to serve as a general network architecture for multi-resolution visual recognition.

**Limitations and Future Work.** The dislocation between the underlying optical resolution of native face images and that of a certain branch may limit the power of the model, which may be improved by selecting the optimal processing branch for the input in combination with the image quality, rather than by image size alone. The optimal branch selection strategy is not fully investigated though we have provided an intuitive way to select the branch for inputs (see Figure 6). Importantly, based on the unified multi-resolution metric space, the underlying resolution of the inputs (integrated spatial resolution with quality assessment) can be utilized to provide the reliability of the representation and contribute to risk-controlled face recognition. They will be our future research directions.

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
