# OpenReview forum: "Learning Unified Representations for Multi-Resolution Face Recognition"
_NeurIPS.cc/2022/Conference — NeurIPS 2022 Submitted_

### Official Review · Reviewer_NEKY · 2022-07-11

**Rating:** 5
**Confidence:** 3
**Soundness:** 3 good
**Presentation:** 2 fair
**Contribution:** 2 fair

**Summary:**

This paper proposes a representation learning method, called Branch-to-Trunk network (BTNet), for multi-resolution face recognition. It consists of a trunk network (TNet) given in the form of a unified encoder, and multiple branch networks (BNets) that are used for resolution adaptation. A resolution-specific BNet is used on the input and the output are implanted as feature maps in the feature pyramid of TNet, at a layer with the same resolution. Using branch distillation and backward-compatible training, BTNet transfers high-resolution information to multiple branches, while keeping representation compatibility. Experiments show effective performance on face recognition benchmarks, both for multi-resolution identity matching and feature aggregation.

**Questions:**

Did the authors perform experiments on the branch selection process? Even preliminary tests would be useful.

A figure reporting on overview of the proposed network it would have been useful.

See also weaknesses above.

**Limitations:**

Limitations have been discussed by the authors.

No ethical issues have been reported.


**Strengths And Weaknesses:**

Strengths
- Code has been included in the submission supplementary files.
- Experiments have been provided on several datasets.

Weaknesses
The presentation of the results should be improved, Authors should better comment the different cases and propose explanation for specific situations where results are not intuitive in the tables.

The selection strategy for the optimal branch is not fully investigated. Author acknowledged this while exposing the limitations of their work, but I think this is something very relevant to the method that required more in depth understanding.

Performance outperform state-of-the-art on one dataset, while this does not occur on others. Authors should report motivations on why this happens. Which are the dataset characteristics that best explain this fact?

The paper writing needs some improvements. There is also some typos to check. See for example:
- page 3, line 111: "(3) This also effectively reduce" --> reduces

============= POST REBUTTAL ================
The rebuttal partially answered to my questions. I prefer to change my score to borderline accept.

---

> ### Author Response · Authors · 2022-08-02
> **Response to Reviewer NEKY**
>
> We would like to thank the reviewer for the careful reading and valuable comments. We address the questions and clarify the issues accordingly as described below.
>
> **Q1: The presentation of the results**
>
> **[Reply]** Many thanks for your suggestions. We will add more detailed explanation, especially for the counter-intuitive results in the revised paper.
>
> **Q2: The branch selection strategy**
>
> **[Reply]** We performed some preliminary tests on the strategy and found that the low-quality image may possess an underlying optical resolution significantly lower than its size due to degraded quality caused by noise, blur, occlusion, etc. Thus, there exists dislocation between the underlying optical resolution of native face images and that of a branch. To avoid introducing extra large-scale parameters for predicting the image quality, the heuristic selection strategy is used in our paper. We agree that this requires more in depth understanding and will investigate more in the future.
>
> **Q3: SOTA performance on QMUL-SurvFace**
>
> **[Reply]** For the six verification benchmarks (e.g., LFW, AgeDB-30) and IJB-C dataset, there is no official protocols for validating multi-resolution face recognition and corresponding SOTA methods. Without loss of generality, we build new protocols to validate the relative gain of our method to baselines. Different from the above datasets, QMUL-SurvFace is dedicated to surveillance face recognition challenge, which has standard testing protocols and reported SOTA method. The wide spatial resolution distribution of QMUL-SurvFace enables a more comprehensive evaluation of the performance on multi-resolution face recognition.
>
> **Q4: Paper writing**
>
> **[Reply]** Many thanks for your advice. We will improve the paper writing, carefully check and correct typos in the revised paper.
>
> **Q5: Overview figure**
>
> **[Reply]** An overview of the proposed network is illustrated in Figure 2 in the paper, which shows the basic ideas as well. For the detailed architecture of an instantiation of our network, please refer to Figure 1 in Appendix.

---

### Official Review · Reviewer_RGyV · 2022-07-11

**Rating:** 3
**Confidence:** 5
**Soundness:** 2 fair
**Presentation:** 3 good
**Contribution:** 2 fair

**Summary:**

In this paper, the authors introduce a Branch-to-Trunk Network (BTNet) for multi-resolution Face Recognition.
Particularly, BTNet consists of two main components: BNet and TNet. While TNet aims at maintaining the main features flow (from high- to low-resolution), BNet is introduced for maps inputs to intermediate features with same resolution and maintain the feature compatibility when injecting them to the main flow of TNet.
The proposed BTNet is validated on QMUL-SurvFace and IJB-C protocols.

**Questions:**

1. The improvement achieved by BTNet is unclear:
BTNet aims at training TNet for the main embedding flow, BNet aims at adopting the feature map extracted from the direct low-res images for making its features compatible to the feature flow of TNet. Moreover, only L_cur is adopted for the feature discriminative as [3].
How can BTNet achieve improvements in comparison to prior works? How can it leave a large gap in comparison to phi_{mr} when same L_cur is adopted?

2. Some baselines are missing:
- In most experiments, phi_{hr}, phi_{mm}, and phi_{mr} are employed. However, in my opinion, these settings are quite “naive”, i.e. phi_{mr} is simply augmentation of multi-resolution images with similar weights for both HR and LR images.
They are acceptable for some basic baselines. The authors are recommended to have more baselines such as lower weight for LR images during training or enforce the features of LR images closer to the HR ones.
Then, the baselines would be more complete.

3. In Table 3, why is the accuracy of phi_{mm} higher than phi_{mr} in cross-resolution experiments? How about the compatibility of phi_{mm} in cross-resolution recognition?

4. The comparisons in Table 3, 4 are lacking of prior works on IBJ-C.

**Limitations:**

Yes

**Strengths And Weaknesses:**

STRENGTH
- The paper is well-motivated.
- The writing is easy to follow.
- The idea of multi-branch network to overcome the limitation of the amount of information can be extracted from low-resolution input faces is interesting.
- Experimental results show improvements in comparison to baselines and prior works.


WEAKNESS
Generally, the novelty of the paper is limited as most of the building blocks (i.e. loss from CurricularFace [3] and common distillation loss) are introduced in previous works.
Although the idea of a multi-branch network to overcome the robustness against low-resolution images, there are several concerns about this design. Please see the Question Section for further comments and questions.

---

> ### Author Response · Authors · 2022-08-02
> **Response to Reviewer RGyV**
>
> We would like to thank the reviewer for the careful reading and valuable comments. We address the questions and clarify the issues accordingly as described below.
>
> **Q0: Technical novelty of our paper**
>
> **[Reply]** We agree that most of the building blocks were introduced in previous works. However, the key technical contribution of the paper is a general network structure based on Branch-to-Trunk and effective training strategies based on BCT and branch distillation. Since multi-resolution face recognition is dominated by super-resolution and projection methods, our method is the first attempt to decouple the information flow conditioned on the input resolution. Therefore, we believe our method is technically novel.
>
> **Q1: Analysis of improvement achieved by our method**
>
> **[Reply]** Actually, $L_{cur}$ is an implementation of $L_{influence}$, which mainly aims at ensuring the feature compatibility instead of improving the discriminability. In our experiments, the same $L_{cur}$ is adopted for BTNet and $\varphi_{mr}$. Though BTNet and $\varphi_{mr}$ both improve the robustness against the resolution-variance, BTNet leaves a large gap in terms of operating schemes. Specifically, instead of rescaling the input images to a canonical scale before learning to be robust against introduced noise, BTNet operates them on different scales separately and then transforms them to a uniform scale. It makes up for the sacrificed discriminability of $\varphi_{mr}$, which is in exchange for the adaptability for resolution-variance.
>
> **Q2: More baselines**
>
> **[Reply]** Many thanks for your suggestions. We agree that $\varphi_{mr}$ seems naive as HR and LR images share similar weights during training and we did some tests for other alternatives. To fairly compare with the baselines, we trained the models in the same settings. For $\varphi_{mr}$, we trained the model with a size in the candidate set [7,14,28,56,112] with equal probability of being chosen. For baseline A, we chose in the candidate set [7,14,28,56,112] with unequal probability of being chosen [0.3 0.25 0.2 0.15 0.1]. For baseline B, we randomly down-sampled the images to [4,112]. For baseline C, we finetuned $\varphi_{hr}$ by using L2-loss to enforce the features of LR images closer to the HR ones. Higher weight for LR images (Baseline A) results in worse performance as it is difficult for the model to learn to extract discriminative features from a limited number of HR images; lower weight for LR images (Baseline B) still leads to limited robustness against low-resolution images. Directly enforcing the features closer in the output feature space (Baseline C) would affect the supervision from the class label, making the discriminative power of the model almost lost. We see our design is critical to achieving higher accuracy. We will add more complete baselines in our revised paper.
>
> ||112&14 Acc.(%)|14&14 Acc.(%)|
> |:-|:-|:-|
> |Baseline A| 87.13  | 80.22      |
> |Baseline B|88.13|80.55|
> |Baseline C|  -    |-|
> |Ours|94.08|90.90|
>
> **Q3: Analysis of cross-resolution experiments**
>
> **[Reply]** In the experiments corresponding to Table 3, the cross-resolution features are concatenated to form the representation of the identity, i.e., one person ID. Though the features within an identity are not compatible, the components of the embeddings are still decoupled, and thus the similarity calculation, i.e., the dot product of two vectors is operated on different components respectively. From the experimental result that $\varphi_{hr}$ performs well, we can view the HR component as sufficiently discriminative information while the LR component as weak interference information. Since $\varphi_{mm}$ can extract more informative HR features than $\varphi_{mr}$, it is reasonable to have higher accuracy. As $\varphi_{mm}$ can extract more informative LR features than $\varphi_{mr}$, the interference is much more intense for $\varphi_{mr}$ with the resolution decrease of the LR component (i.e., from 112 to 7), and thus the gap is becoming larger.
>
> **Q4: Comparison on IJB-C**
>
> **[Reply]**  In fact, our proposed BTNet is a general network architecture for representing multi-resolution information that can improve the multi-resolution representation capability of recent methods (e.g., more powerful hierarchical DNNs with improved marginal losses).  We built a test benchmark for multi-resolution feature aggregation by modifying the official one of IJB-C dataset. It aims to verify the relative improvement to existing face recognition methods brought by our operating scheme, rather than the optimality of the absolute metrics.

---

> > ### Comment · Reviewer_RGyV · 2022-08-09
> > **Further Concerns**
> >
> > I appreciate the authors for the responses to my concerns. I still have some further concerns regarding to the response.
> >
> > **Q3.** It is unclear about the setup of $\varphi_{mm}$.
> > As $\varphi_{r}$ has the issue of representation compatibility, it is not suitable for cross-resolution recognition (i.e. see Lines 258-259 of the submission). Table 1 also confirms the lower accuracy of $\varphi_{mm}$ in comparison to other methods.
> > However, in Table 3, $\varphi_{mm}$ is better than $\varphi_{mr}$. Is there any inconsistency in the benchmarks?
> >
> > **Q4**. I understand the authors' viewpoint. However, I believe the comparison to SOTA approaches should be employed so that we can see the potential of using the proposed approach in practice.
> > From my viewpoint, the results are not necessary to beat all SOTA approaches. A comparable result with an advantage of multi-resolution robustness is also a valuable point.

---

### Official Review · Reviewer_ra6W · 2022-07-16

**Rating:** 4
**Confidence:** 5
**Soundness:** 3 good
**Presentation:** 3 good
**Contribution:** 3 good

**Summary:**

The paper proposes a Branch-to-Trunk network (BTNet) with multiple independent branch networks (BNets) and a shared trunk network (TNet) to extract feature representation for multi-resolution face recognition. The CurricularFace loss is refined as influence loss and a branch distillation loss is included for training to ensure the discriminative and compatible representation. The experiments conducted on both 1:1 and 1:N verification seems to validate the efficacy of the proposed method.

**Questions:**

See the above "weakness".

**Limitations:**

Yes.

**Strengths And Weaknesses:**

### Strengths:

- The paper is well organized and easy to follow.

- It is an interesting idea to decouple a face image as branches for discriminative representation learning and then couple them as the trunk for compatible representation learning.

- The performance on the tasks of both cross-resolution and same-resolution identify matching looks very promising.


### Weaknesses:

- The authors claim the proposed BTNet is able to learn unified representation for face recognition. However, the experiment validation is only conducted on the face verification task. Will the learned unified representation can be beneficial to solve the general face recognition tasks?

- The comparisons is inadequate to well demonstrate the superiority of the proposed BTNet.  All the baselines listed in Table 2 are at least two years ago.  The authors should compare the proposed BTNet with some recent face identification methods.

- The necessary ablation studies are missing. In the current experimental results presentation, we only can see the overall performance gain. It is hard to guess which parts contribute more, and which parts contribute less. If the main performance gain is due to the introduction of CurricularFace loss as influence loss, then main technical contribution of TBNet may need to be doubted.

- Lack the necessary qualitative results to show both successful and failure cases.

---

> ### Author Response · Authors · 2022-08-02
> **Response to Reviewer ra6W**
>
> We would like to thank the reviewer for the careful reading and valuable comments. We address the questions and clarify the issues accordingly as described below.
>
> **Q1: General face recognition tasks**
>
> **[Reply]** Generally, The testing protocols of face recognition can be categorized into (1) face verification: the model needs to output whether the input images are of the same person or the input image is that of the claimed person (1:1 problem); and (2) face identification: the model needs to output the ID if the image is any of the K persons from a given database (1：N problem). We evaluate not only face verification task on six verification benchmarks (e.g., LFW, AgeDB-30) and IJB-C dataset, but also face identification task on IJB-C and QMUL-SurvFace datasets. Results show that our method performs well consistently in above two tasks.
>
> **Q2: Comparison on QMUL-SurvFace**
>
> **[Reply]** In fact, our proposed BTNet is a general network architecture for representing multi-resolution information that can improve the multi-resolution representation capability of recent methods (e.g., more powerful hierarchical DNNs with improved marginal losses). We show that though AdaFace [a] with additional subtle strategies performs better than the best method reported in [b] but underperforms our method. As stated above, we can implement BTNet based on AdaFace and obtain more gains potentially. We will add more complete results in the revised paper.
>
> |TPIR20(%)@FPIR|0.3|0.2|0.1|0.01|AUC|
> |:-|:-|:-|:-|:-|:-|
> |RAN [b]|26.5|21.6|14.9|3.8|32.3|
> |AdaFace [a]|28.3|23.6|16.5|2.6|32.6|
> |Ours|31.2|26.9|20.6|2.5|35.4|
>
> **Q3: Explanation for influence loss**
>
> **[Reply]** Introducing the influence loss is mainly aimed to ensure the representation compatibility, and any classification loss can be used for implementation. The comparison results demonstrate that there is no significant difference among different implementations of influence loss. Moreover, in Appendix A.3, we did the ablation study on the effects of different training method alternatives, showing how each strategy (e.g., back-compatible training, branch distillation) contributes to the effectiveness of BTNet.
>
> |Implementation of influence loss|112&14 Acc.(%)|14&14 Acc.(%)|
> |:-|:-|:-|
> |CosFace|94.10|90.78|
> |ArcFace|94.17|90.88|
> |CurricularFace|94.08|90.90|
>
> **Q4: Qualitative results of successful and failure cases**
>
> **[Reply]** Thanks for your advice. Since we cannot post images in the OpenReview system, we will add visualization and analysis of both successful and failure cases in the revised paper.
>
> [a] AdaFace: Quality Adaptive Margin for Face Recognition
>
> [b] Generate to Adapt: Resolution Adaption Network for Surveillance Face Recognition

---

### Author Response · Authors · 2022-08-02
**A Brief Recap of Our Work**

Thanks for all reviewers’ careful and valuable comments. To better understand our work, we would like to give a brief recap here.

**(1) What is our goal?**  Matching images with arbitrary resolutions (i.e., high-resolution, cross-resolution and low-resolution) effectively and efficiently, which is quite different from the traditional face recognition task.

**(2) What is the core idea of our method?** Building unified (i.e.,compatible and discriminative) representations for multi-resolution images without introducing erroneous information.

**(3) How to achieve our goal via our method?**

|                                | Compatibility                                         | Discriminability                           |
| :-                            |:-                                                             |:-                                                   |
|Input preprocessing|-                                                              |w/o rescaling to a canonical size |
|Network strcture     |TNet (unified encoder)                             |BNets (resolution adapters)         |
|Training strategy    |Back compatible training (influence loss)|Branch distillation                        |

---

### Meta-Review · Area_Chair_JAx5 · 2022-08-26

**Recommendation:** Reject
**Confidence:** Certain

**Metareview:**

This paper proposes a Branch-to-Trunk network with multiple independent branch networks and a shared trunk network for multi-resolution face recognition. This paper received three detailed reviews.  While there are some merits in this work, the reviewers raised many concerns, including  1) inadequate experiments to demonstrate the superiority of the proposed method, 2) missing ablations, 3) improvement achieved by BTNet is unclear.  After reading the reviews, rebuttals, and the paper, the AC concurs with the reviewers’ comments, and feels that the concerns outweigh the strength. Therefore, a rejection is recommended.

**Award:**

No

---

### Decision · Program_Chairs · 2022-09-14

Reject